# Using single remote sensing image to calculate the height of the landslide dam and the maximum volume of the lake

Weijie Zou [1,2], Yi Zhou [1], Shixin Wang[1], Futao Wang[1], Litao Wang[1], Qing Zhao[1], Wenliang Liu[1], Jinfeng Zhu[1], Yibing Xiong [1,2], Zhenqing Wang[1,2], Gang Qin [1,2]

[1]*Aerospace Information Research Institute, Chinese Academy of Sciences, Beijing, 100094, China;*
[2]*University of Chinese Academy of Sciences, Beijing 100049, China;*

*Correspondence: Yi Zhou (zhouyi@radi.ac.cn) and Futao Wang (wangft@aircas.ac.cn)*

# 1. Abstract

Landslide dams are caused by landslide materials blocking rivers. After the occurrence of large-scale landslides, it is necessary to conduct large-scale investigation of barrier lakes and rapid risk assessment. Remote sensing is an important means to achieve this goal. However, at present remote sensing is only used for monitoring and extraction of hydrological parameters at present, without prediction on potential hazard of the landslide dam. The key parameters of the barrier dam, such as the dam height and the maximum volume, still need to be obtained based on field investigation, which is time-consuming. Our research proposes a procedure that is able to calculate the height of the landslide dam and the maximum volume of the barrier lake, using single remote sensing image and pre-landslide DEM. The procedure includes four modules: (a) determining the elevation of the lake level, (b) determining the elevation of the bottom of the dam, (c) calculating the highest height of the dam, (d) predicting the lowest crest height of the dam and the maximum volume. Finally, the sensitivity analysis of the parameters during the procedure and the analysis of the influence of different resolution images is carried out. This procedure is mainly demonstrated through Baige landslide dam and Hongshiyan landslide dam. The single remote sensing image and pre-landslide DEM are used to predict the height of the dam and the key parameters of the dam break, which are in good agreement with the measured data. This procedure can effectively support the quick decision-making regarding hazard mitigation.

Keywords: Landslide dam, Remote sensing, DEM, Dam height, Hazard

# 2. Introduction

Landslide dams are caused by landslide materials blocking rivers, usually in mountainous areas with rivers and narrow valleys, bringing great risks to local people's lives and property(Costa and Schuster, 1988; Fan et al., 2020). Landslide dams disaster is widely distributed around the world. For instance, the 11 dams caused by the Magnitude 7.6 earthquake in New Zealand 1929(Adams, 1981); Oso Landslide Dam in Washington, USA in 2014(Iverson et al., 2015); Diexi Landslide Dam on Minjiang River, China, 1933(Li et al., 1986); Yigong Landslide Dam in 2000(Zhou et al., 2016) and a series of landslide dams including the Tangjiashan Landslide Dam caused by the Wenchuan earthquake in 2008(Zhang et al., 2019).Based on the historical records of 183 landslide dams, Costa found that the main way of dam breaching was overtopping. 41% of dams breached within one week, and 85% breached within a year(Costa and Schuster, 1988). Respectively Fan analyzed a series of dams induced by the 2008 Wenchuan earthquake finding that 43% of them collapsed within one month(Fan et al., 2012). And according to Shen's research on the longevity of the barrier lake, nearly 48.3% of the dams will breach within a week, and 84.4% of the dams will fail within one year(Shen et al., 2020). Most of landslide dams are unstable. However, the landslide dam always occurred in remote mountainous areas, with inconvenient traffic conditions and poor infrastructure(Cui et al., 2009). When earthquakes or precipitation induce large-scale landslides, field survey is time-consuming and manpower-consuming(Dong et al., 2014). Remote areas tend to be more vulnerable and the dam breaching are more likely to cause serious consequences. So, it requires us to identify the landslide dam and take action as quickly as possible.

There are several factors influencing the process of formation, development and risk of landslide dams. These factors can be divided into three categories. First, the factor of the soil, including the dam material composition and the repose angle of the dam material, has an unavoidable relationship with the formation and erosion process of the dan. The low permeability and high erodibility will lead to short longevity of the landslide dam and fast breaching of the dam(Shen et al., 2020). Second, the hydrological parameters, such as lake volume, average annual discharge and catchment area which decide the speed of lake surface raising(Cao et al., 2011). The faster the lake raises, the less time is left to hazard mitigation. Third, the geometric parameters, such as the length and angle of the landslide surface and the length, width, height of the dam. The landslide surface influences the kinetic energy of the landslide material which has a great influence on the formation of the landslide dam. And the geometric parameters of the dam itself decide the stability of dam, the maximum volume of the lake and the potential maximum discharge of breaching (Dong et al., 2011a; Cao et al., 2011; Shen et al., 2020).

Remote sensing has the ability to identify and monitor landslide dams on a large scale conveniently, and supports quick decision-making regarding hazard mitigation(Canuti et al., 2004; Fan et al., 2021). In the research before, remote sensing is usually regarded as an auxiliary means to monitor the change of the catchment area or to measure the length of the dam. For example, Wang and Lv used multiple remote sensing images to extract water boundary images and pre-landslide DEM to monitor the changes of lake

volume of Yigong Lake(Wang and Lu, 2002). Respectively, Cheng et al. proposed a method to estimate reservoir capacity of water based on water boundary and DEM(Chen and Lu, 2008).

The research above focused on obtaining information about the barrier lake through remote sensing and Geographic Information System. However, these kinds of methods focus on monitoring and can only obtain part of geometry parameters directly through image such as catchment area. Some essential components of hazard evaluation are not available in these research. Especially the height of the dam which determines the maximum volume of the barrier lake and the flood peak of the dam breaching(Costa and Schuster, 1988; Ermini and Casagli, 2003; Peng and Zhang, 2012; Dong et al., 2014) cannot be obtained through these methods. However, as most of the landslide dams breach by overtop, they start to breach as long as the elevation of lake surface equals the elevation of the landslide dam(Meng et al., 2021; Costa and Schuster, 1988; Ermini and Casagli, 2003). So, the height of the landslide dam decides the maximum volume of the lake. The damage of the landslide dam mostly relies on the flood it causes through breaching. As water goes through the dam surface, the erosion process will lead to rapid increase of the discharge and finally result in flood. According to research, his process has a strong relationship with the height of the landslide dam(Anon, 2021; Shen et al., 2020; Chen et al., 2004; Braun et al., 2018), which makes it one of the most important parameters related to this hazard.

With the rapid development of Unmanned Aerial Vehicles (UAVs), in 2008, photogrammetric UAVs are also used to survey the landslide dams in the Wenchuan earthquake in 2008(Cui et al., 2009). However, after the earthquake, there are to be a large number of landslides and the affected area is considerably huge. If UAVs are used for precise investigation one by one, it cannot meet the requirements of timeliness for the emergency. Based on the pre-landslide DTM and a series of remote sensing images after the landslide dam, Dong obtains the variation of the lake level to estimate the slope foot of the barrier dam and predict the dam height, completing a quick assessment of the dam breaching hazard(Dong et al., 2014). But this procedure is still inconvenient as it requires sequential images to predict the height of the dam. All of the methods that use the pre-landslide DEM are based on an important assumption that the pre-landslide DEM is reliable. Nevertheless, take Baige Landslide Dam as an example (Fig 1), we can find that the elevation of landslide area changes greatly. The landslide area has a greater degree of subsidence, and the dam area has a greater degree of uplift. And even in areas nearby covered with vegetation, there was about 20 meters of subsidence averagely, which demonstrates that the assumption above nee further improvement.

This research will focus on the weakness above using single remote sensing image and pre-landslide DEM to obtain the essential information of the landslide dam and calculating the height of the landslide dam based on the formation mechanism of the landslide dam. The Baige Landslide Dam is taken as an example to verify the feasibility of this procedure. And the sensitivity analysis of the parameters during the procedure and the analysis of the influence of different image resolution will be carried out in the "discussion" part.

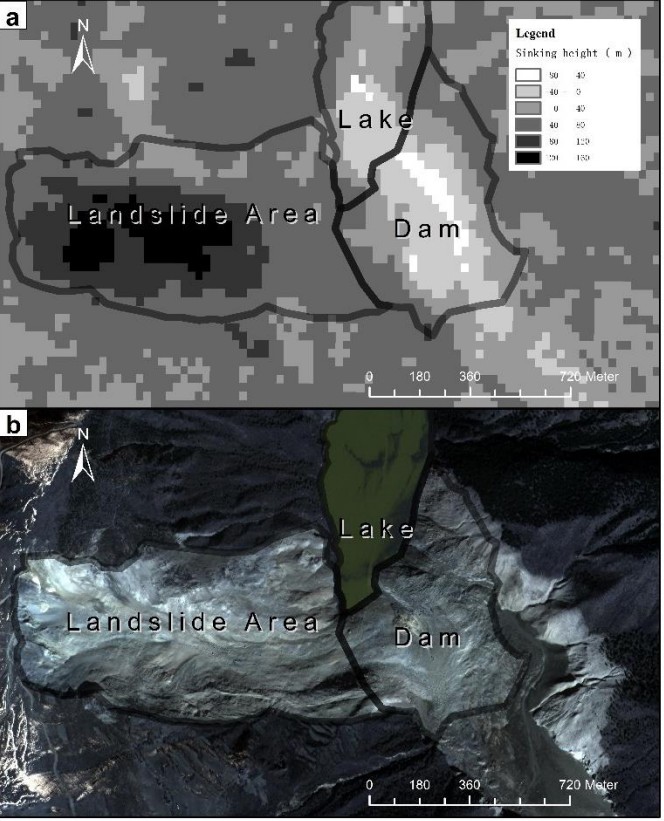

Fig 1 picture a is the comparation of pre-landslide DEM (SRTM V3) and the after-landslide DEM. And picture b is the remote sensing image from Beijing-1 satellite (taken in November 9, 2018)

# 3. Procedure

After the occurrence of large-scale landslides, the government often can't get all the disaster situation immediately, so large-scale landslides investigation is needed. As the disaster often occurs in remote areas, the purpose of the large-scale investigation is not only to find the landslide dams, but also to make an objective evaluation of the hazard of the landslide dams, supporting reasonable allocation of resources to avoid excessive reaction. When a landslide dam is identified from the image, the procedure to calculate its height is divided into four parts: (a) selecting the reference points to determine the elevation of the lake level; (b) estimating the elevation of the bottom of the dam; (c) calculating the highest elevation of the dam crest based on the formation mechanism of the landslide dam; (d) predicting the lowest height of the dam crest and the maximum of the lake volume. This section will elaborate the details of (a), (b), (c) and (d), obtaining the lowest height of the dam crest and calculating the maximum volume based on GIS.

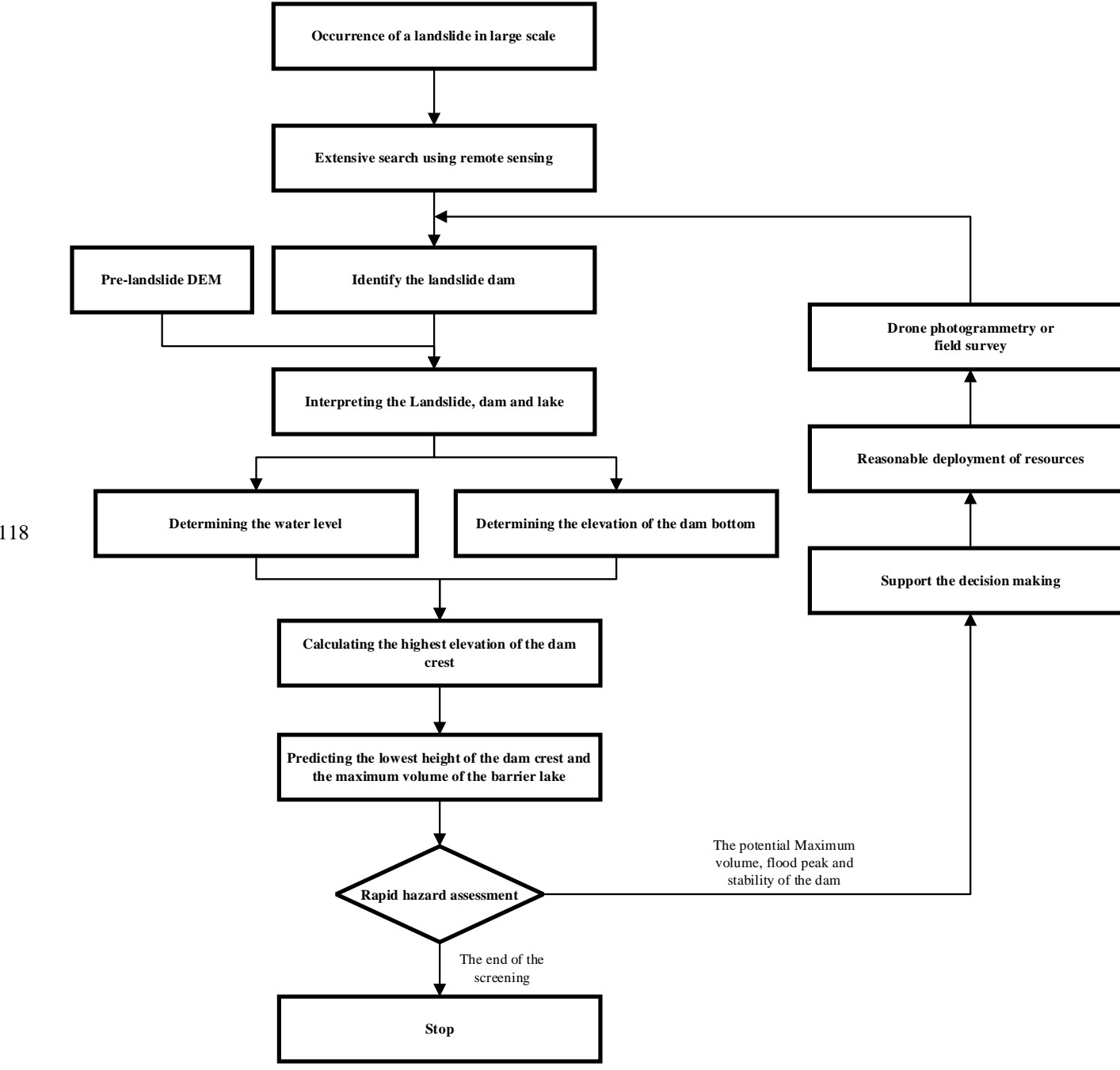


Fig 2 the procedure of obtaining the height of the dam crest and completing the hazard assessment

## 3.1. Determining the elevation of the lake level

The method of estimating the elevation of the barrier lake based on remote sensing images has been
practiced by many scholars. Typically speaking, researchers assume that the elevation of the water
boundary is the same as the topography. And pre-landslide DEM is used in most cases to determine the
lake level with the water boundary in the image(Wang and Lu, 2002; Chen and Lu, 2008; Dong et al.,
2014; Braun et al., 2018). However, the reliability of the pre-landslide DEM may decrease as a result of
landslides (Fig 1). The reasons are summarized as follows: (a) the landslide has caused some changes in
the topography of the area; (b) the pre-landslide DEM has errors itself, especially in the mountainous
area; (c) as the pre-landslide DEM usually cannot be undated in time, there can be some landslides
without records before.
For the reasons above, the selection of the reference points to determine the elevation of the lake level
should follow these principles to reduce errors. (a) As landslides often bring about large-scale ground
subsidence, when selecting reference points, the point around the landslide area should be avoided. (b)
Because landslides and settlements tend to occur in areas with steep terrain and little vegetation
coverage(Ayalew and Yamagishi, 2005) and the DEM is more precise in flat terrain, the reference points
should be in vegetation-covered flat terrain, avoiding gully or ravines.
Under these strictions the reference points selected can be regarded as having the same elevation of the
lake level. Therefore, the lake level is determined. However, in order to determine the elevation of the
lake level, a complex number of reference points are needed. Their value can't be the same for the random
errors but should be within a certain range (Fig 7, Fig 9), for the random errors of DEM and the errors in
the process of determining the points. In this situation, points that are one and a half interquartile range
away from the mean value are considered outliers. And the elevation of the lake level is the average
elevation of the remains. Because the dam blocks the channel and the river has no outflow, the water
surface can be assumed to be still(Wang and Lu, 2002; Morgenstern et al., 2021; Fan et al., 2021). So,
the elevation of the lake level is the same as the elevation of the lake-dam point in Fig 3.

## 3.2. Determining the elevation of the dam bottom

In this procedure, the bottom of the dam refers to the point where the dam meets the river bed on the
downstream side. In practical cases, the most reliable method is to directly use the riverbed elevation
obtained recently. In the absence of relevant data, the following method should be taken for prediction.
Within a certain range, the riverbed elevation can be considered to decrease in proportion along the
channel, conforming to a linear variation. Therefore, sampling elevation points at the lowest point of the
river valley in the pre-landslide DEM, removing the outliers and carrying out simple regression to obtain
the fitting of the riverbed elevation. By extending the fitting results to the dam body and subtracting the
historical river depth, the bottom elevation of the dam is obtained.
However, the historical river depth is to vary with seasons. So, there must be some errors in this prediction.
The influence of dam bottom elevation on calculating dam height will be analyzed in the "discussion"
section.

## 3.3. Calculating the highest elevation of the dam crest

According to Wu's laboratory experimental study, the geometrical form of the barrier dam is mainly
determined by landslide slope, river slope, angle of repose, earthwork amount and sliding height.    (Wu
et al., 2020).
With his theory, if the river is completely blocked and the valley can be simplified into U-shape, the
longitudinal section of the landslide dam can be simplified as a trapezoid(Wu et al., 2020) as shown in
Fig 3. And the trapezoid will follow the following pattern.

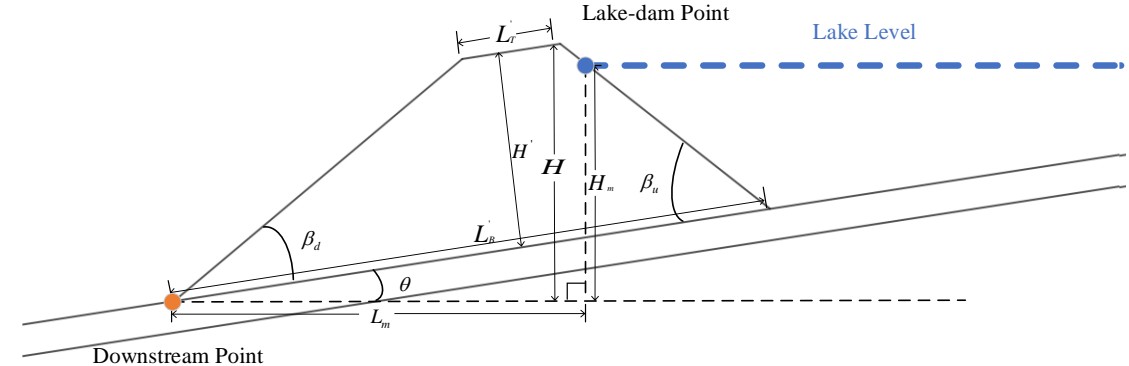

Fig 3 simplified section of the landslide dam
The top of the dam is parallel to the bottom of the dam (Wu et al., 2020).
$L_T^{'} // L_B^{'}$ (1)
Where $L_T^{'}$ is the top of the dam, $L_B^{'}$ is the bottom of the dam (Wu et al., 2020).
$\beta_d + \theta = \beta_u - \theta = \chi\varphi$ (2)
Where $\beta_d$ is the angle between the body of the dam and the riverbed on the downstream side, $\beta_u$ is
the angle between the body of the dam and the riverbed on the upstream side, $\varphi$ is the angle of repose
of the landslide mass and $\chi$ is the parameter that fits the effect of "cut top" phenomenon. $\varphi$ is
determined by the nature of the soil itself and $\chi$ will be affected by landslide surface angle, landslide
length and other factors(Grasselli et al., 2000). The determining of the $\chi$ can be simplified as
follows(Wu et al., 2020):
$\chi = 0.57 + 0.51(1 + e^{\frac{(\alpha-34)}{10.50}})^{-1}$ (3)
where $\alpha$ is the angle of the landslide surface. As the angle is higher, the actual angle between the
riverbed and the landslide material will be smaller and the length of the dam along the river will be longer.
Normally speaking, this formula fits the actual situation well. The precise of this fitting will be discussed
in the "discussion" section.
According to Wang's field investigation on the Wenchuan earthquake, it is found that the angle of repose
of landslide dam in the Wenchuan earthquake is between 28.8° and 44.7°, with an average of 35.5°(Wang
et al., 2013). In the absence of relevant data, it is recommended to use the average provided by Wang.
$\varphi = 35.5°$ (4)
Wu proposed that the height of the dam has a certain relationship with the length of the bottom of the
dam (Wu et al., 2020), as follows:
$$H^{'} = (0.37 + 1.1\tan\theta)\cdot\tan(\beta_d + \theta)\cdot L_B^{'} \quad (5)$$
where $H^{'}$ is the height between the dam top and the dam bottom, $\theta$ is the angle of the riverbed and
$L_B^{'}$ is the length of the dam along the river. The $R^2$ of formula (1) (2) (3) (5) are all greater than 0.95.
As shown in Fig 3, the elevation of the dam-lake point and the elevation of the dam bottom has already
been obtained before. So, $H_m$ can be calculated and $L_m$ can be obtained directly from the remote
sensing images. According to formula (1), (2), (3), (4) and (5), using simple geometric relations, the
following relation can be obtained:

$$L_B^{'} = \frac{L_m}{\cos\theta} + \frac{\cos(\beta_u - \theta)}{\sin\beta_u}\cdot(H_m - L_m\cdot\tan\theta) \quad (6)$$
$$H_r = \sin\theta\cdot(L_B^{'} - H^{'}\cdot\tan\theta - H^{'}\cdot\tan(90 - \beta_u)) \quad (7)$$

$$H = \frac{H^{'}}{\cos\theta} + H_r \quad (8)$$
Where $H$ is the difference between the highest elevation of the dam crest and the dam bottom
elevation and $H_r$ is the difference of the elevation of the riverbed between the dam bottom and the
crest. $\theta$ and $\alpha$ can be obtained through the remote sensing image and the pre-landslide DEM easily.
Through this procedure, the highest elevation of dam crest is determined based on a single image and
pre-landslide DEM, which can be used in the further prediction of the dam breaching and related
decision-making.

## 3.4. Predicting the lowest height of the dam crest and the

## maximum volume of the barrier lake

Because the height of the landslide dam in the vertical direction of the river channel will not be
consistent(Costa and Schuster, 1988; Fan et al., 2020), but will form different types of distribution
according to the characteristics of the case, resulting in the height of the landslide dam is not a simple
value but a range. As the most important factor affecting the dam breaching is the height of the lowest
point of the dam crest, which determines the potential maximum volume of the barrier lake and the
maximum discharge volume of the dam breach(Costa and Schuster, 1988; Chen et al., 2004, 2021; Dong
et al., 2011b, 2014; Yang et al., 2013; Zhong et al., 2018), the prediction result of the highest elevation
of the dam crest can't be used in related breaching models directly.
But by simply analyzing the highest elevation of the dam crest and the lowest elevation in the existing
records, a simple estimation of the relationship between them is carried out, as shown in Fig 4.

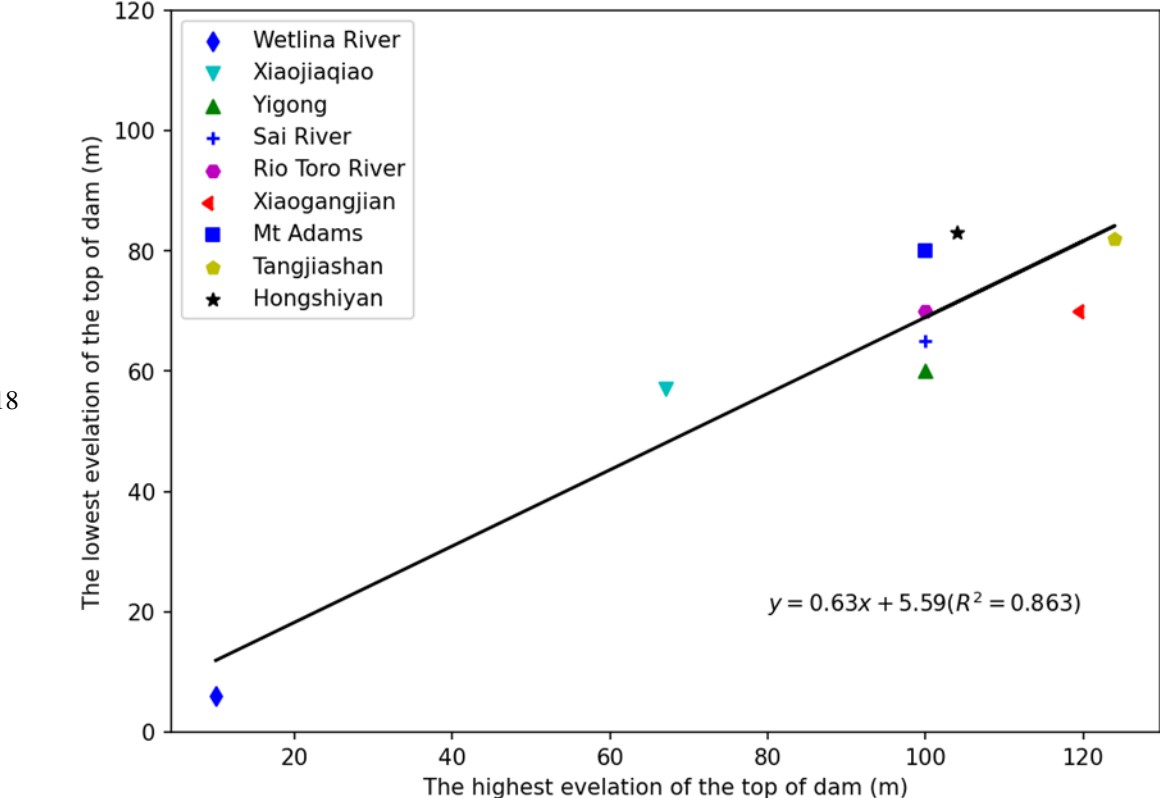

219  Fig 4 the relationship between the highest elevation of the dam crest and the lowest elevation of the

220  dam crest. These datas can be found in the papers of Cui, Costa, Mora and so on(Costa and Schuster,

221  1991; Mora Castro, 1993; Briaud, 2008; Cui et al., 2009; Peng and Zhang, 2012; Chen et al., 2020).

222  .

223  The relationship can be expressed as follows:

224  $H_l = 0.63H_h + 5.59(R^2 = 0.863)$ (9)

225  where $H_l$ is the lowest elevation of the dam crest and $H_h$ is the highest elevation of the dam crest.

226  On the basis of the formula above, we can use the lowest elevation of the dam crest to complete the rapid

227  assessment of the breaching hazard.

228

# 4. Validation of the proposed procedure

## 4.1. Baige Landslide Dam

231  The Jinsha River, the upper reach of the Yangtze River, was dammed twice recently at Baige, Tibet, one

232  on 10 October 2018 and the other on 3 November 2018 (UTC+8), at 98°42′32.24″E, 31°4′59.27″N(Fig

5) (Zhang et al., 2019) and one on November 3, 2018, the residual landslide of "10.10" Baige Landslide
Dam slid down again, forming "11.03" Baige Landslide Dam on the basis of the original residual dam(Li
et al., 2019). The dam is much larger than the first one, as the width of the dam top is 195 m, the length
of the dam top is 273 m and the highest elevation of the dam crest is 3014m(Chen et al., 2020). After
proper treatment, its storage capacity is reduced from $8.69 \times 10^8 m^3$ to $5.79 \times 10^8 m^3$ and the flood
peak is diminished from $41624\, m^3/s$ to $31000\, m^3/s$ (Chen et al., 2020; Yunjian et al., 2021). A
large number of roads and bridges were damaged downstream, and a total of 54,000 people were affected,
with economic loss of over 7.43 billion yuan(Zhang et al., 2019). Due to abundant field survey data and
its great harm, Baige Landslide Dam was selected to demonstrate this procedure.
Baige Landslide Dam occurred in a deep valley of the mountainous area and the barrier lake is long and
narrow (Fig 6). In order to demonstrate the proposed procedure, the second Baige landslide is taken as
example. The image used is a 0.8m resolution image from Beijing-1 which was taken on November 9,
2018 and the pre-landslide DEM we choose is SRTM V3 of 30m resolution which was taken in 2000.
The effect of the resolution of the image will be discussed in the "Discussion" section.

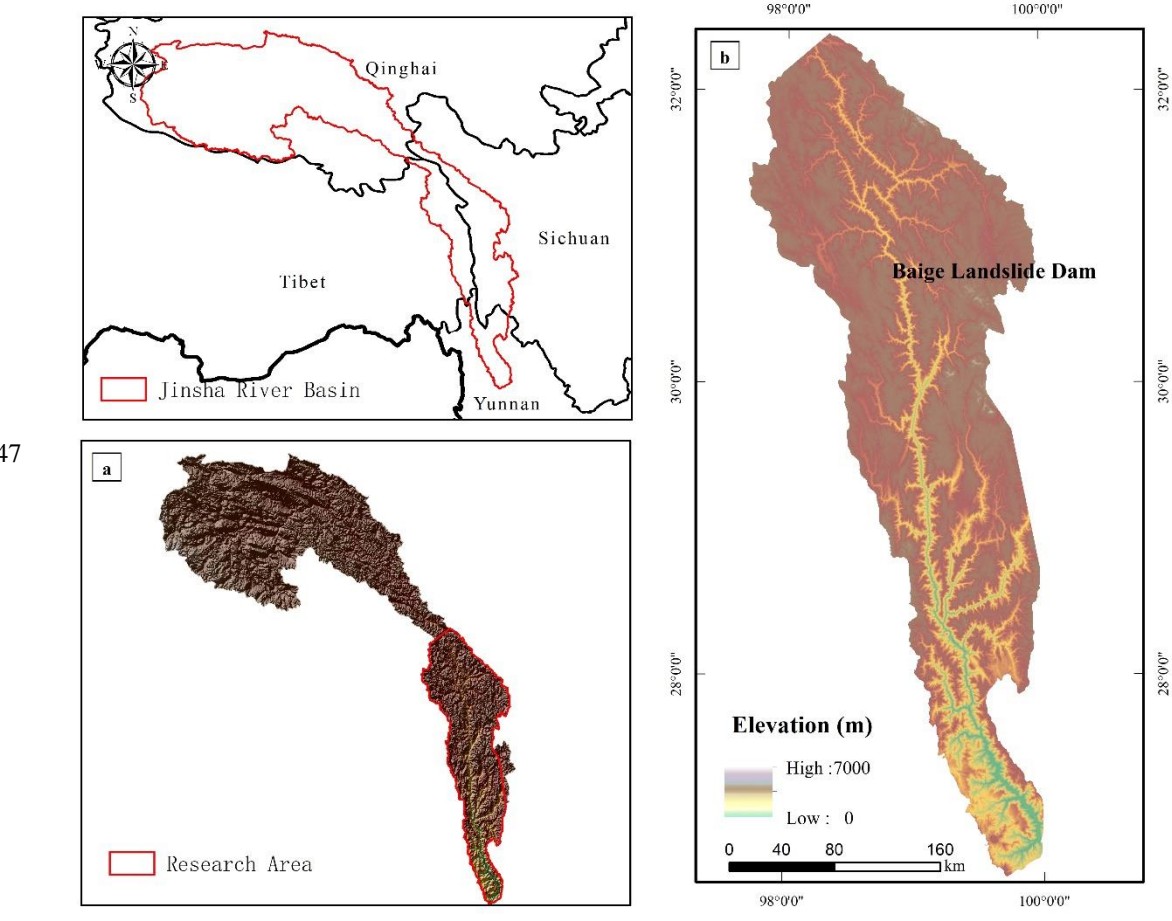

248                            Fig 5 the position of the Baige Landslide Dam

**Determine the elevation of the lake level**
At the water boundary in the remote sensing image, the area covered by vegetation with relatively flat
terrain and a certain distance from the landslide was selected for elevation sampling (Fig 6). Under ideal
circumstances, the distribution of sampling points' elevation should be completely consistent. But in

practice, there are often large deviations, shown in Fig 7, the specific reasons for which have been discussed in the "Procedure" section and will not be repeated. The deviation between the maximum and minimum elevation of sampling points can reach 72m, and the shape basically conforms to the normal distribution. Therefore, the mean of reference points can be obtained directly after clearing the outliers, which is the elevation of barrier lake and the outcome is 2944m. Since the lake is essentially still, the elevation of the lake should be the same as the elevation of the point where the dam meets the lake, shown as the triangle in Fig 3.

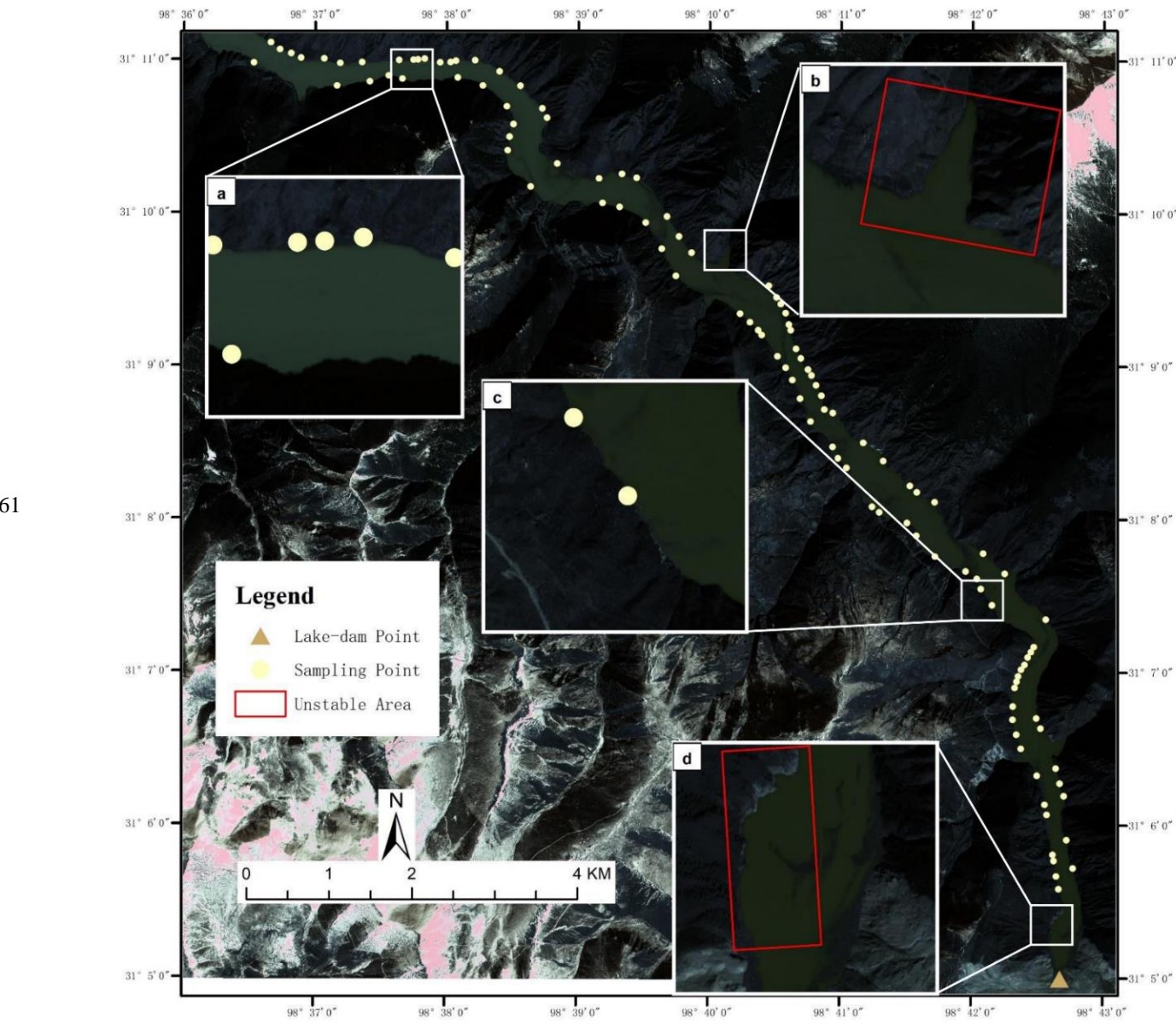

Fig 6 the sampling points in the case of Baige Landslide Dam (image from Beijing-1 satellite)

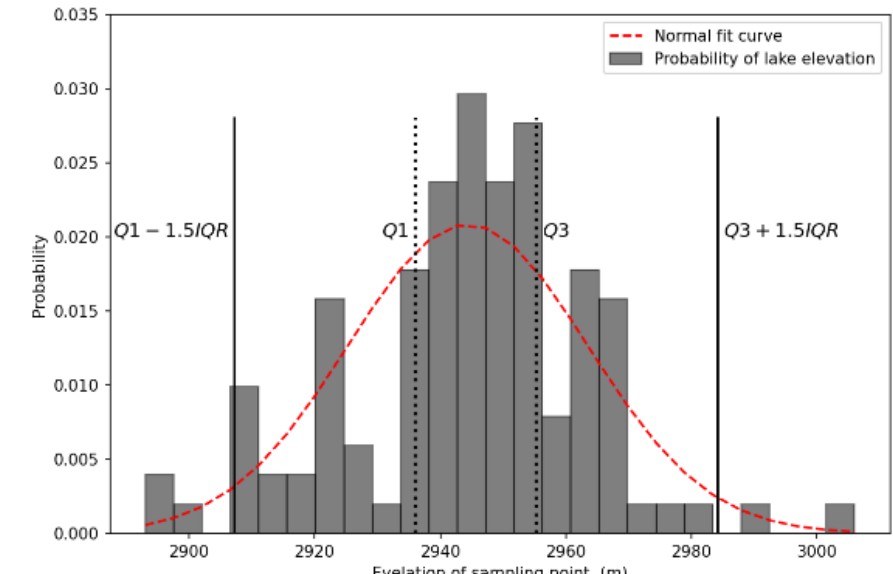

Fig 7 elevation distribution of sampling points in the case of Baige landslide dam

The Intersection over Union (IOU) of the area with elevation below 2944m in DEM and the actual submerged area in the remote sensing image is 84.48% (Fig 8). The two are found to be basically consistent.

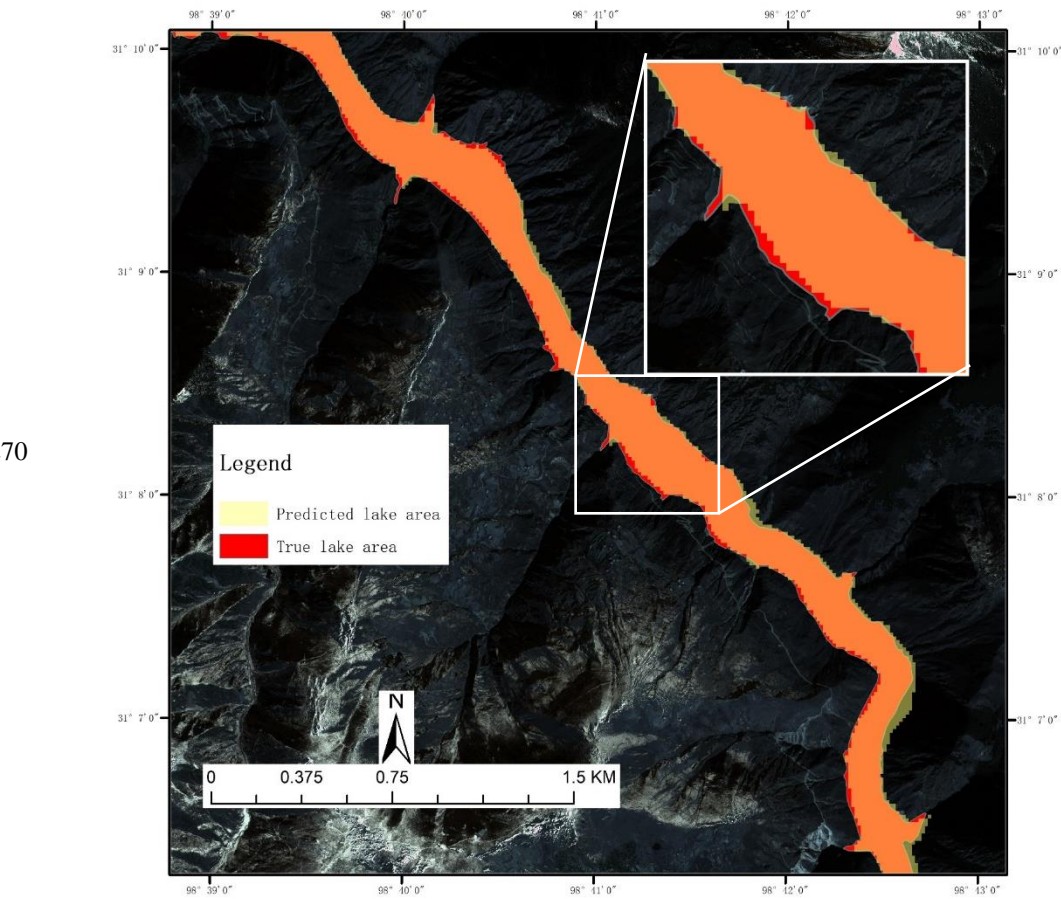

Fig 8 the comparation of the area with elevation below 2944m in DEM and the actual submerged area

272    in the remote sensing image(image from Beijing-1 satellite)

**Determining the elevation of the dam bottom**

The inclination angle of the riverbed is calculated by sampling and unitary regression and is about 0.11°. The elevation of the water level on the place of dam bottom before the landslide is 2867m. As the water depth is not considered when obtaining DEM and varies with change of rainfall in the rainy season and dry season, this value can't be used directly. According to the date in China Ministry of Water Resources Information Center, the water depth of Jinsha River section is about 2-10m. The water depth can be assumed as the mean value, 6m. Therefore, the final estimate of the dam bottom elevation is 2861m. Respectively, according to the field survey, the riverbed elevation is 2860m(Chen et al., 2020).

**Calculating the highest height of the dam crest**

The slope angle of the landslide surface, the inclination angle of the riverbed and the length of the landslide can be calculated directly through remote sensing image and DEM. The slope angle of landslide surface is 30.65°. The inclination angle of the riverbed is 0.11°. And the length of the landslide that can be observed is 567m. According to formula (5) (6) (7) (8), with the parameters obtained before, the highest height of the dam top is 155.4m and the highest elevation of the dam top is 3016.5m with an error of 2.5m compared to the measured data by Chen, 3014m(Chen et al., 2020).

**Predicting the lowest height of the dam crest and the maximum volume of the barrier lake**

Taking Baige Landslide Dam as an example, according to the case section, we have predicted that the highest elevation of the dam crest is 3016.5m and the height of the dam is 155.4m. According to formula (9), the lowest height of the crest of the landslide dam is 104.2m and the elevation is 2964.2m with an error of 2.8m compared to the measured data by Chen, 2067m(Chen et al., 2020). Using Geographic Information System, we can estimate based on DEM(Wang and Lu, 2002; Chen and Lu, 2008) that its potential maximum volume is $7.96 \times 10^8 (m^3)$.

## 4.2. Hongshiyan landslide dam

Another case for validation is Hongshiyan landslide dam, a landslide created by moderate earthquake (Ms 6.5) on August 3$^{rd}$, 2014. The epicenter of the earthquake is located at 27.11° N, 103.35° E and the landslide is 8.8 km southeast from the epicenter(Luo et al., 2019). The landslide dam is holding a maximum water storage of $2.6 \times 10^8 (m^3)$ (Zhou et al., 2015). Breaching of this giant dam will not only pose a high threat to the residents who live around, but also bring a possibility to damage other hydropower dams downstream. The data used to carry out the procedure in this research and predict the

essential geometry parameter of landslide dam is listed in Table 1, including an after-landslide remote
sensing image (2 m solution) and a pre-event DEM (30 m solution).

| Input data | Source | Description |
|---|---|---|
| After-landslide Remote sensing image | Gaofen-1 satellite | 2 m solution |
| Pre-landslide DEM | SRTM V3 | 30 m solution |
| Repose angle of the debris | Relative case recording | Rough estimation |
| The elevation of riverbed | Sampling from DEM | Rough estimation |

Table 1 Source of input data used in Hongshiyan landslide dam case.

**Determine the elevation of the lake level**

The image and the DEM are used to obtain the parameters required to make the prediction. The elevation
of the lake level is obtained by sampling lake edge points. The distribution of the sampling points is
shown in the Fig 9 and the elevation of the lake level is 1170 m.

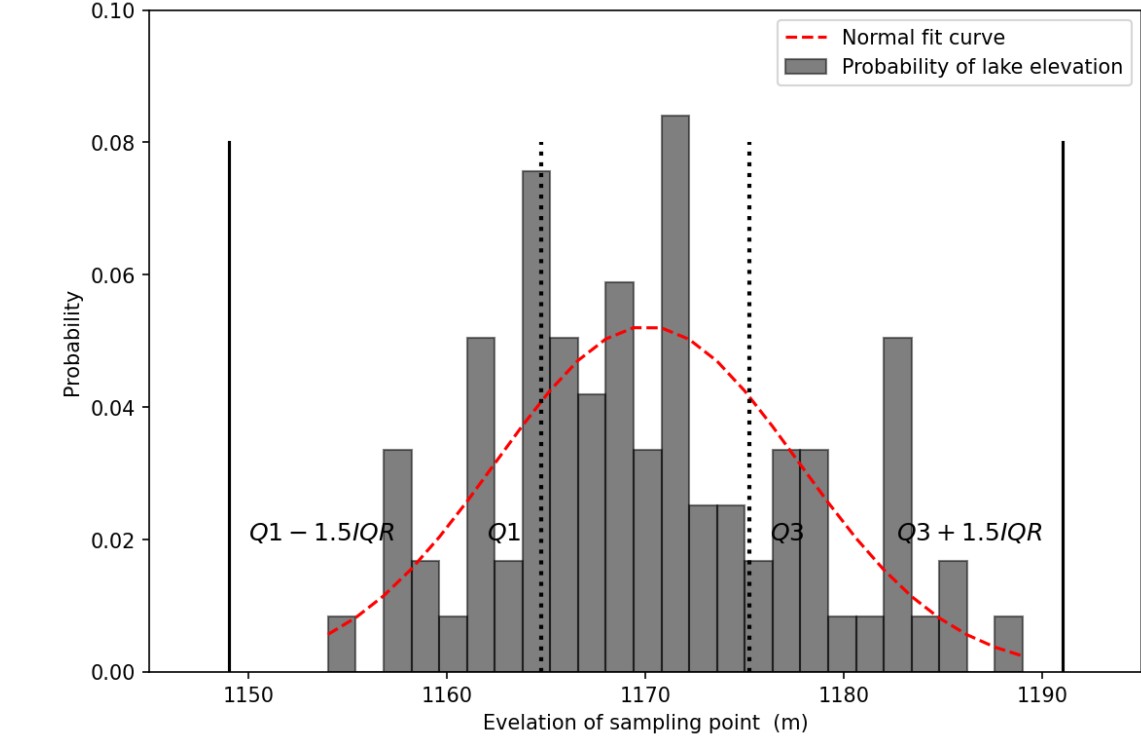

Fig 9 elevation distribution of sampling points in the case of Hongshiyan landslide dam

**Determining the elevation of the dam bottom**

As shown in Fig 10, the pre-event elevation of the water level on the place of dam bottom can be obtained through sampling the lowest points along the riverbed in the DEM, which is 1114m. As the water depth of River is about 3 m(Zhou et al., 2015), the elevation of the dam bottom is 1111m.

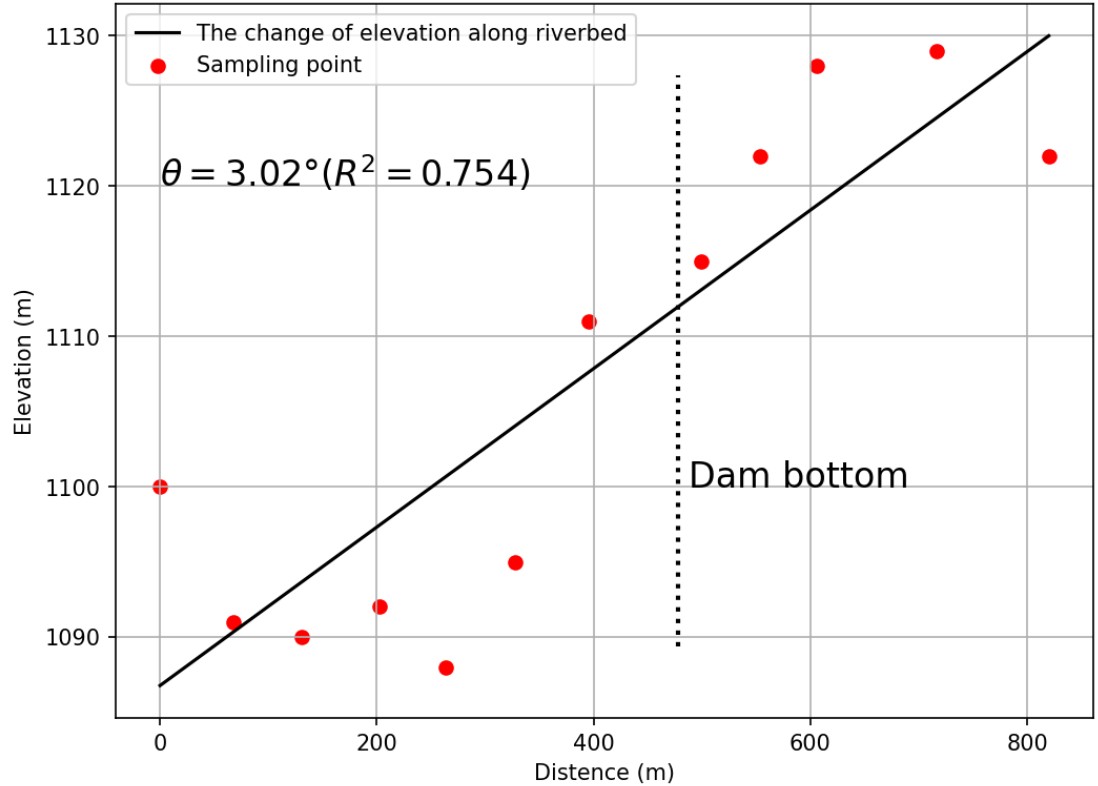

Fig 10 the elevation changes along the riverbed in the case of Hongshiyan landslide dam

**Calculating the highest height of the dam crest**

Landslide dams are caused by landslide materials blocking rivers. After the occurrence of large-scale landslides, it is necessary to conduct large-scale investigation of barrier lakes and rapid risk assessment. Remote sensing is an important means to achieve this goal. However, at present remote sensing is only used for monitoring and extraction of hydrological parameters at present, without prediction on potential hazard of the landslide dam. The key parameters of the barrier dam, such as the dam height and the maximum volume, still need to be obtained based on field investigation, which is time-consuming. Our research proposes a procedure that is able to calculate the height of the landslide dam and the maximum volume of the barrier lake, using single remote sensing image and pre-landslide DEM. The procedure includes four modules: (a) determining the elevation of the lake level, (b) determining the elevation of the bottom of the dam, (c) calculating the highest height of the dam, (d) predicting the lowest crest height of the dam and the maximum volume. Finally, the sensitivity analysis of the parameters during the

procedure and the analysis of the influence of different resolution images is carried out. This procedure is mainly demonstrated through Baige landslide dam and Hongshiyan landslide dam. The single remote sensing image and pre-landslide DEM are used to predict the height of the dam and the key parameters of the dam break, which are in good agreement with the measured data. This procedure can effectively support the quick decision-making regarding hazard mitigation.

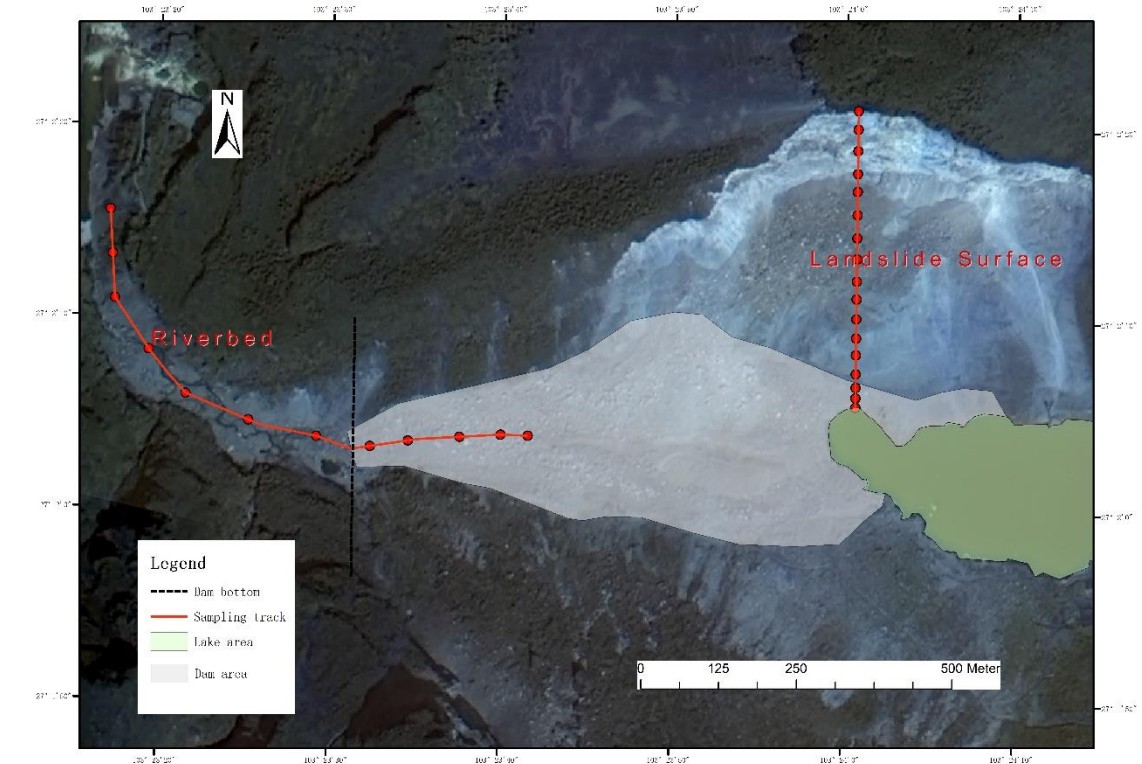

Fig 11 Hongshiyan landslide dam (image from Gaofen -1 satellite)

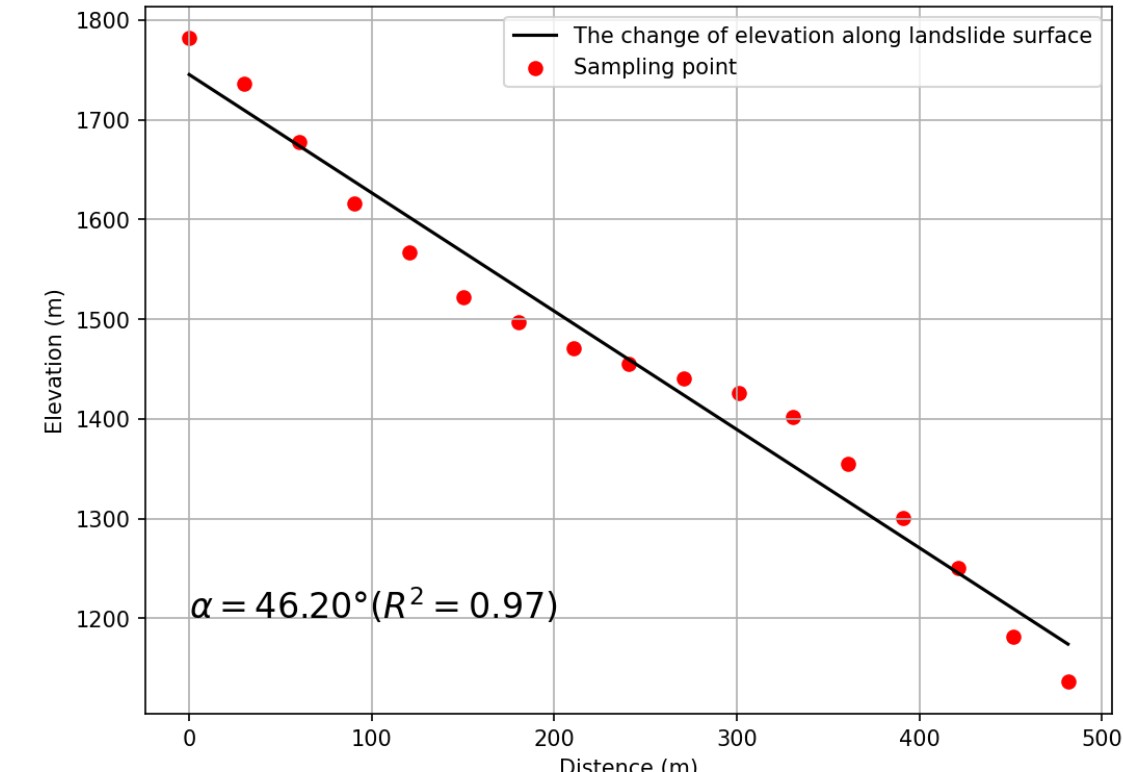

346        Fig 12 the elevation changes cross the landslide surface in the case of Hongshiyan landslide dam


The length of the landslide dam that can be observed, $L_m$, is measured directly in the image (Fig 11),
which is 737.4 m. Angle of the riverbed $\theta$ which is 3.02° (Fig 10) and the landslide surface $\alpha$ which
is 46.20° (Fig 12) can be calculated through analysis of the changes of the elevation along the river and
the landslide. As the recording of the repose angle of the debris is missing, the average value of other
cases is taken as a rough estimation. According to the average value of other landslide dam(Wang et al.,
2013)., it is set as 35.5°.
Putting the parameters above into the formula (6) (7) (8), we can calculate the highest elevation of the
dam crest, which is 1269.9m.
**Predicting the lowest height of the dam crest and the maximum volume of the barrier lake**
As it is the lowest elevation of the dam crest that decides the break of dam, formula (9) is used to fitting
the relationship between the lowest crest and the highest crest. The elevation of the lowest elevation of
the dam crest is 1216.7 m. And the potential maximum volume of the lake can be calculated easily with
the DEM. The comparison of field survey and predicting outcome is shown in Table 2, which suggests a
strong consistency between them.

| Parameter | Measured data | The predicting outcome | Error |
|---|---|---|---|
| the lowest elevation of the dam top | $1222\,(m)$ | $1216.7\,(m)$ | $5.3\,(m)$ |

| the maximum of lake volume | $2.60 \times 10^8 (m^3)^*$ | $3.09 \times 10^8 (m^3)$ | $0.49 \times 10^8 (m^3)^*$ |

Table 2 comparation between predicting outcome and measured data from field survey(Zhou et al., 2015; Luo et al., 2019)

# 5. Discussion

## 5.1. Rapid hazard assessment

The lowest height of the dam crest and the maximum volume of the barrier lake are important input parameters for the dam-breaking model. This paper has given the procedure to obtain them rapidly. We take Baige landslide dam as an example to illustrate how to use the prediction results to carry out rapidly hazard assessment.

Many scholars have found the correlation between the geometric parameters of landslide dam and its risk by empirical formula. On the basis of the prediction results and the formulas they provide, we can make a quick prediction of the key information of the landslide dam hazard, such as the dam volume, the stability of the barrier dam and the potential maximum discharge of the lake.

**Predicting volume of the dam**

The width of the barrier dam can be obtained directly from remote sensing images, which is $574.6m$. As the edge and Angle conditions in the simplified model(Fig 4) have been cleared, that is, all the simplified section plane parameters in the model can be obtained. So based on the relationship between edges and angles in the model, the distance between top and bottom in the lowest crest, $H_l^{'}$, and the length of the dam top, $L_T^{'}$, can be expressed by the following formula (10), (11).

$$H_l^{'} = \cos\theta(0.63H_h + 5.59 - H_r)\,(10)$$

$$L_T^{'} = L_B^{'} - \frac{H^{'}}{\tan\beta_d} - \frac{H^{'}}{\tan\beta_u}\,(11)$$

However, because the cross section of the barrier dam is not evenly distributed in the direction of the vertical river, the height change will occur as discussed in 3.5. We can assume that the change of its top height is basically linear and the bottom side length and top side length of the section trapezoid do not change in the direction of the vertical channel. Therefore, we can obtain the estimation Formula (12) to

calculate the volume of the dam debris. In the case of Baige landslide dam, the prediction outcome is $32.4 \times 10^6 m^3$, and the true value according to field survey is $30.2 \times 10^6 m^3$ (Shen et al., 2020). The error is mainly induced by the elevation change of riverbed in the direction of the vertical channel., which has a great influence to area of the dam section when the width of the dam is large.

$$V_d = \frac{1}{4} W(H_l' + H_h')(L_B' + L_T')$$ (12)

**Predicting the stability of the landslide dam**

In Dong research, a regression model to evaluate the stability of the barrier lake is proposed based on the case of the historical landslide dam(Dong et al., 2011a), as shown in Formula (13).

$$L_s = -2.55 \log(P) - 3.64 \log(H_l) + 2.99 \log(W) + 2.73(L) - 3.87$$ (13)

where $P, H_l, W, L$ are the inflow, dam height, width and length of the landslide dam. In the case of Baige landslide. The inflow of Baige landslide dam is $822 m^3 / s$ (Li et al., 2019). The result $L_s$ is -1.472, which means that Baige landslide dam is unstable and has a high risk to breach.

**Predicting the peak discharge of the breaching**

In the simple prediction formula (14) proposed by Cenderelli., V is the maximum volume of the dammed lake, and Q is the maximum flood peak of dam breaching. Without treatment, the largest flood peak of the Baige Landslide Dam breaching will reach $42257 \, (m^3 / s)$.

$$Q = 3.4 \cdot V^{0.46}$$ (14)

The comparison between the predicted result and the measured date, as shown in table 3, achieves a good agreement. The rapid assessment of the dam breaching hazard has been completed. As the simulation model of dam breaching has a significant influence on the prediction of these factors, they should also be selected carefully in practical applications. Besides formulas above, there are also many other formulas to choose to complete the prediction(Costa and Schuster, 1991; Walder and OConnor, 1997; Shi et al., 2014; Ruan et al., 2021; Peng and Zhang, 2012; Zhong et al., 2018; Ermini and Casagli, 2003; Dong et al., 2011a; Shen et al., 2020). And many scholars have discussed the merits and demerits between these hazard assessment models(Peng and Zhang, 2012; Fan et al., 2021).

| Parameter | Measured data | The predicting outcome |
| --- | --- | --- |
| The highest elevation of the dam top | $3014 \, (m)$ | $3016.5 \, (m)$ |

| | | |
|---|---|---|
| The lowest elevation of the dam top | $2967\,(m)$ | $2964.2(m)$ |
| The maximum of lake volume | $8.69\times10^{8}(m^{3})^{*}$ | $7.96\times10^{8}(m^{3})$ |
| The dam volume | $30.2\times10^{6}(m^{3})$ | $32.4\times10^{6}(m^{3})$ |
| The stability of dam | Not stable | Not stable |
| The peak discharge | $41624\,(m^{3}/s)^{*}$ | $42257\,(m^{3}/s)$ |

Table 3 the comparation of the measured data and the predicted result. As relative measures have been
taken to reduce the maximum volume of the barrier lake, data with star in the table is the estimation
results of Chen's detailed back analyses(Chen et al., 2020).

## 5.2. Sensitivity analysis


This study provides a method to predict critical information about a barrier dam using limited real-time
data. The data required includes an after-landslide satellite image and a pre-event DEM. The data that
is not required include the repose angle of the nearby material and the elevation of the riverbed. If there
are reliable recordings, they can be used in the procedure to improve the prediction accuracy.
Otherwise, our research provides a reliable method to predict them. The process of using of each input
data, determination of intermediate parameters and final output results is shown in Fig 13.

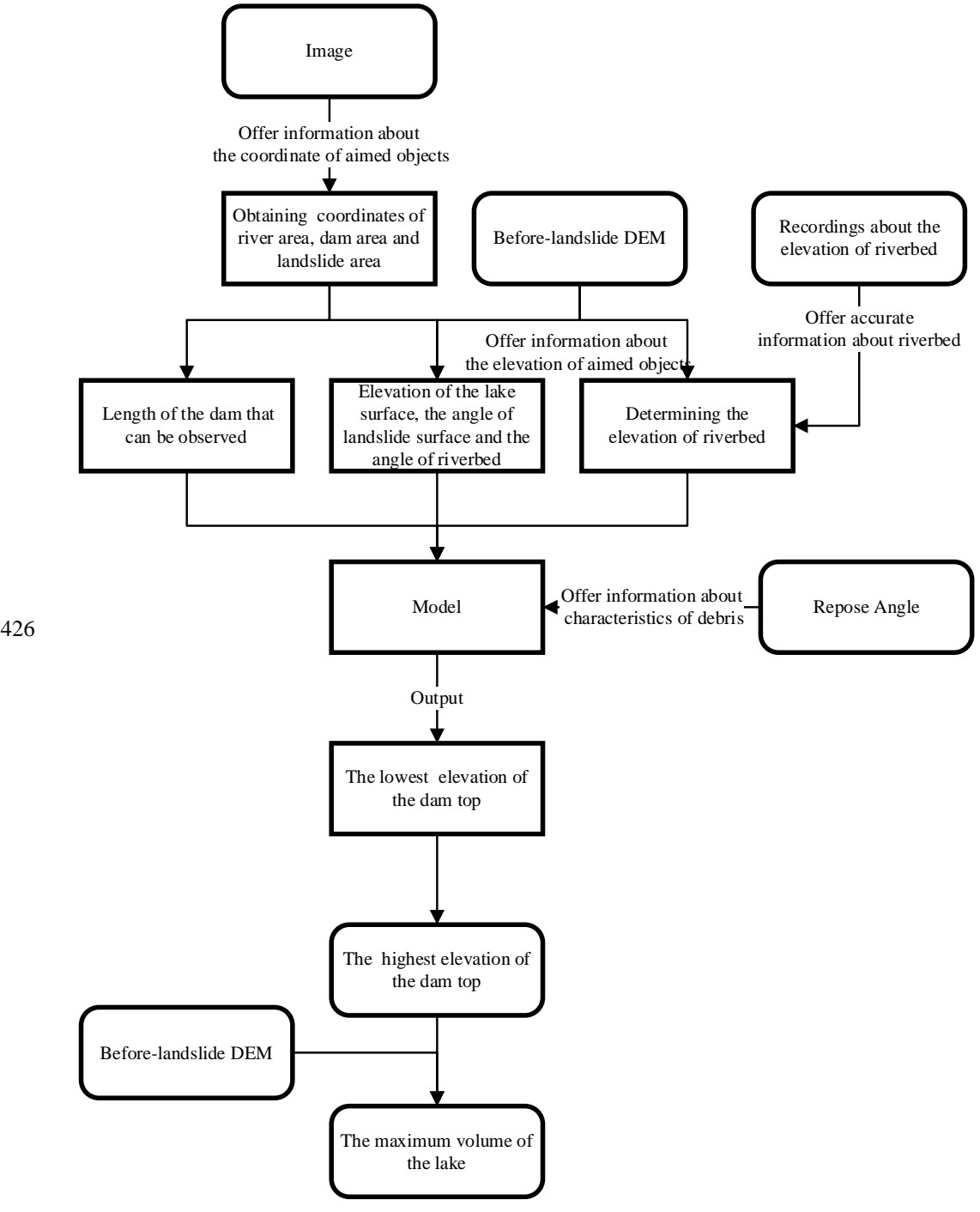

 Fig 13 the complete process of parameters determination


In this procedure, the main parameters put into the model include: the length of the dam that can be
observed, the elevation of the lake level, the elevation of the dam bottom, the slope angle of landslide
surface and the inclination angle of the riverbed. Since $H_m$ is the lake level elevation minus the
elevation of the dam bottom, sensitivity analysis of these two parameters will be conducted on $H_m$
directly.
In order to analyze the sensitivity to these parameters , we take Baige landslide dam as an example. And
the variation of the prediction result with the change of parameters is shown as follow:

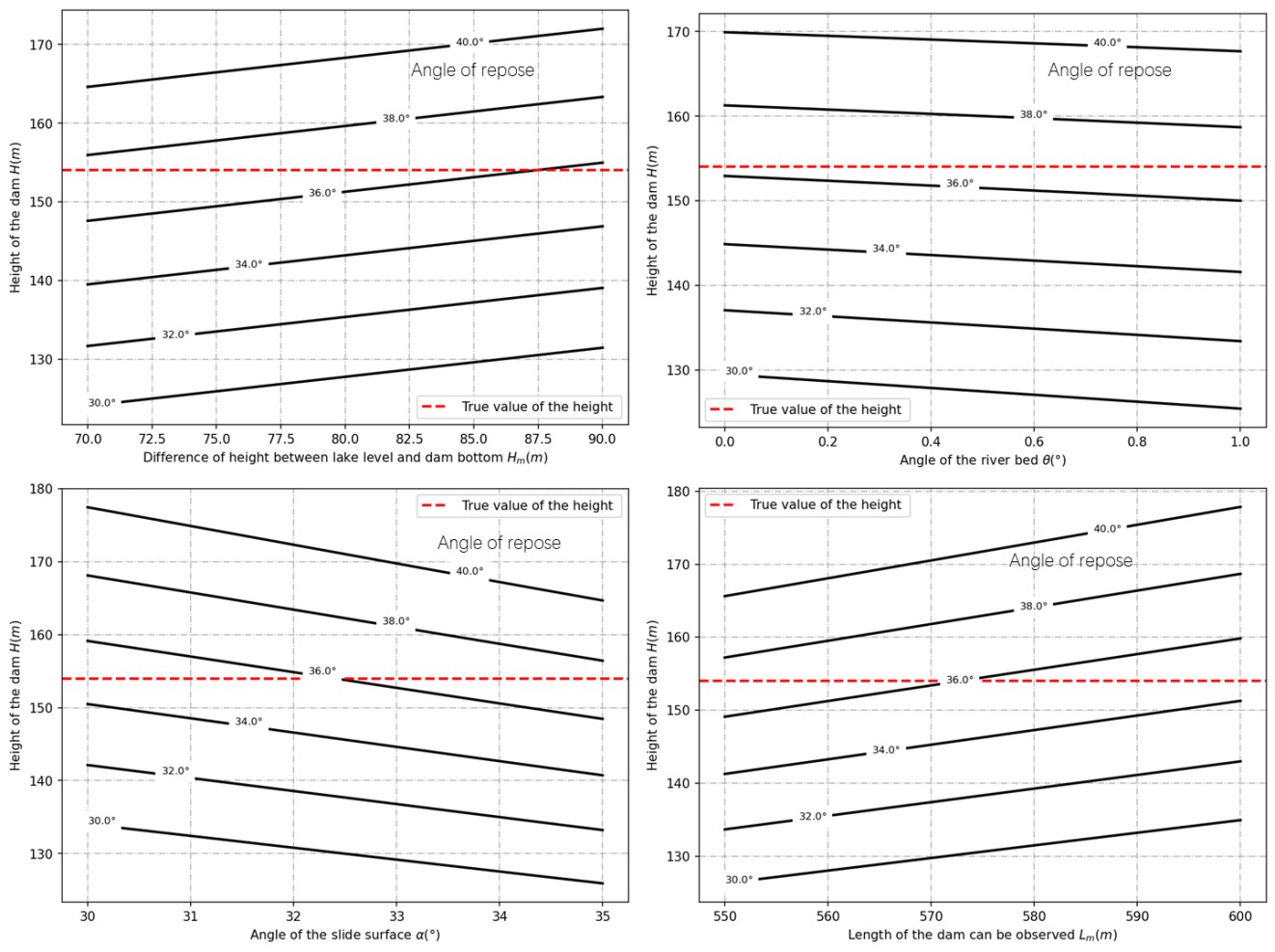


438                    Fig 14 the relationship between the predicted result and the input parameters.


As can be seen from Fig 14, with other parameters unchanged, the greater the observable length of the
dam and the difference of height between the lake level and dam bottom, the higher the dam crest. The
crest of the dam gets lower as the slope angle of landslide surface and the inclination angle of the riverbed
rise. The slope foot of the dam is mainly affected by the angle of landslide surface and inclination angle
of the riverbed. The smaller the slope foot, the smaller the height of the dam. The calculated results are
in good agreement with expectations.
Meantime, it can be found that these parameters all have an impact of about 10% on the final prediction
results. So, it is necessary to be careful to determine these parameters. Possible methods to reduce errors
include repeat procedures and more reliable historical data.

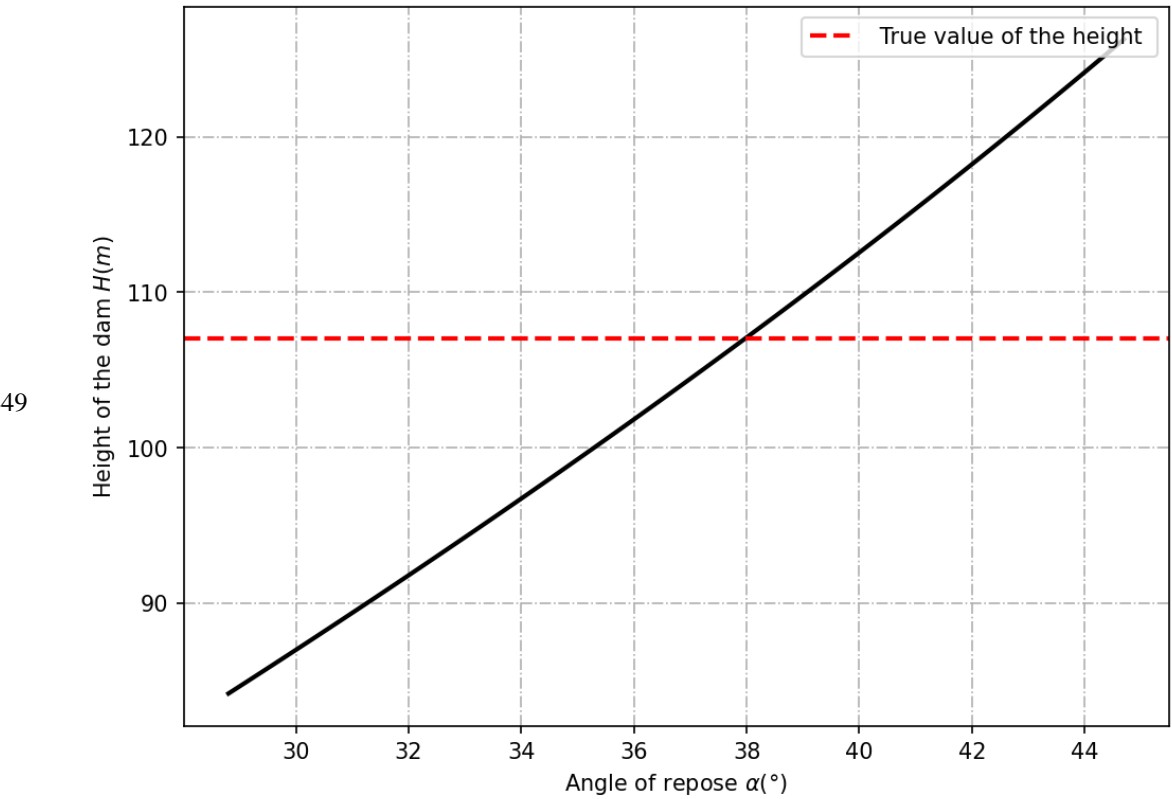

450        Fig 15 the relationship between the predicted result and the angle of repose.


Finally, it is found that the angle of repose of the dam body has a significant influence on the height of
the dam (Fig 15). The greater the angle of repose, the greater the estimate of dam height. According to
Wang's field survey, the angle of repose of the landslide dams in Wenchuan earthquake ranges from 28.8°
to 44.7°, with an average value of 35.5°(Wang et al., 2013). In the absence of the historical date, the
average value proposed by Wang can be used. However, in this way, the difference between the final
result and the true value can be about 30% in the worst case. Therefore, on the premise of sufficient
disaster relief resources, it is better to make a bad estimate of the repose angle, so as not to make a wrong
judgment on the hazard. On the other hand, it is also possible to check the repose angle of the material
in advance in landslide prone area, so as to make a quick hazard assessment after the landslide.
## 5.3. Influence of image solution
The remote sensing image used in the case of Baige landslide dam is Beijing-1 with a resolution of 0.8m
and the pre-landslide DEM is SRTM V3 with a resolution of 30m. As more and more remote sensing
data are available, in addition to satellite-based remote sensing platform, small UAV remote sensing
platform can also be well applied to this procedure. As different sensors and remote sensing platforms
may have different resolutions, we use interpolation to obtain images with different resolutions to explore
the appropriate resolution for this procedure (Table 4, Table 5).

| | Input | | | | | | |
|---|---|---|---|---|---|---|---|
| Resolution | $H_1$ (m) | $H_0$ (m) | $H_m$ (m) | $L_m$ (m) | $\alpha$ (°) | $\theta$ (°) | $\varphi$ (°) |
| 0.8 | 2944 | 2860 | 84 | 567 | 30.65 | 0.11 | 35.5 |
| 5 | 2946 | 2861 | 70 | 545 | 28.58 | 0.10 | 35.5 |
| 15 | 2943 | 2856 | 73 | 562 | 29.44 | 0.09 | 35.5 |
| 30 | 2956 | 2862 | 84 | 540 | 29.10 | 0.16 | 35.5 |

Table 4 the parameters obtained through different resolution image, where $H_1$ is the elevation of the
lake level, $H_0$ is the elevation of the dam bottom, $H_m$ is $H_1$ mines $H_0$, $L_m$ I s the length of the
dam that can be observed in the image, $\alpha$ is the slope angle of landslide surface, $\theta$ is the
inclination angle of the riverbed and $\varphi$ is the angle of repose

| | Output | Accuracy | |
|---|---|---|---|
| Resolution | $H$ (m) | True value $H$ (m) | Error(m) |
| 0.8 | 2964.2 | 2967 | 2.8 |
| 5 | 2964.7 | 2967 | 2.3 |
| 15 | 2961.6 | 2967 | 5.4 |
| 30 | 2960.5 | 2967 | 6.5 |

Table 5 the predicted result of image with different resolutions

As we discussed before, the main parameters in this procedure include the length of the dam that can be
observed, the lake level, the elevation of the dam bottom, the slope angle of landslide surface and the
inclination angle of the riverbed. Obviously, the resolution of the image will affect all of these five (Table
4), but mainly affect the determining of length of the dam that can be observed and the lake level. In
general, the higher the resolution, the more accurate the prediction results obtained. When the resolution
drops from 0.8m to 30m, the error of prediction results changes from 2.8m to 6.5m, as shown in Table 5.
But for the procedure this paper proposed, image with resolution of 5m is sufficient for a good estimate
of the dam height.
There is no doubt that the resolution and quality of DEM data are very important for this procedure.
However, due to the lack of comparative data, this paper does not conduct in-depth discussion on it. For
this part, Dong has had relevant discussions in his research(Dong et al., 2014) for readers' reference.
## 5.4. Other discussion
In this study, the predicting model is mainly based on the formation mechanism of the barrier dam
combined with a single remote sensing image and pre-landslide DEM to quickly predict the essential
paraments of the landslide dam hazard. Therefore, a more comprehensive assessment of the reliability of
formation mechanism has also been carried out. It is found that most laws can be applied well, but
formula (3) has greater limitations in fitting the "cut-top" effect. In Wu's experiment, the "cut-top" effect
fitting is mainly determined by the slope angle of landslide surface. Actually, the angle between the
riverbed plane and slop surface of the dam should be determined by its landslide potential energy,
landslide length, and landslide angle(Grasselli et al., 2000; Xu et al., 2013; Iverson et al., 2015). In
addition to the slope angle of landslide surface, the length of the landslide and potential energy are equally
important. In Wu's formula, only the slope angle of landslide surface is considered, so more experiments
are needed to improve the fitting.
As there are not enough theoretical researches to support the prediction of the lowest elevation of the
dam crest, the method proposed in this paper still has certain limitations. In addition, the mechanism of
the relationship between the highest elevation of the dam crest and the lowest elevation of the dam crest
is not clear. In most cases, when it comes to the height of a barrier lake, usually only the highest or lowest
elevation is recorded, resulting in fewer complete records of both parameters. As the recording in most
cases is not completed, only a small number of cases are used to carry out the fitting. Therefore, this
aspect still needs more work and related researches to support relevant predictions.

# 6. Conclusion

This research proposes a procedure based on a single remote sensing image to predict the height of the
dam crest and rapidly assess the hazard. With the after-landslide remote sensing image, it only takes no
more than one human hour to complete the whole procedure. Compared with Dong's procedure( , this
method only requires only one single remote sensing image and has a wider applicability. In view of the
large topographic changes in the landslide area, a more reasonable method of using the pre-landslide
DEM is proposed. Even the use of poor-quality DEM can complete the relevant prediction and hazard
assessment. In the case of Baige Landslide Dam, by extracting the barrier lake surface elevation and
determining the bottom elevation of the dam, the prediction of the highest elevation of the dam crest is
completed, and the difference between the predicted results and the measured data is within 3m. Since
the lowest point of the dam crest determines the potential maximum volume of the barrier lake, we based
on historical records find that the height of the highest point and the lowest point of the landslide dam
crest basically conforms to the linear relationship. The relationship is expressed as a formula (9) through
unary fitting. The prediction result of the lowest elevation of the crest of the Baige Landslide Dam is
2964.2m, whose error is 2.8m compared to data from field survey, 2967 m. And in the case of Hongshiyan
landslide dam, the error of predicting result of dam top elevation is 5.3m.
In the discussion part, some essential parameters of landslide dam, such as the volume of the dam, the
stability of the dam and the potential maximum flood peak of the dam break without treatment, is
calculated based on the predicting result, which is basically consistent with the true value. The sensitivity
of the parameters used in this method is analyzed, and it is found that the repose angle of the landslide
material can affect the prediction result up to 30%. Therefore, the repose angle should be carefully
determined when using this procedure for related applications. Finally, through experiment with different
resolutions of remote sensing images, we find that as the resolution becomes lower, the accuracy of this

method decreases. The resolution of 5m and above is a reasonable range for applying this method, otherwise it will be difficult to distinguish the dam body and the water boundary.

# Data availability

The data are available from the authors upon request.

# Author Contributions

WJZ designed the experiments, and YZ carried them out. SXW and FTW gave some very important suggestions on basic knowledge of landslide dams. LTW, WLL, ZQ and JFZ helped to operate the whole procedure. QG, ZQW helped with some figures, and YBX provided some remote sensing images. FTW prepared the manuscript with contributions from all co-authors.

# Competing interests

The authors declare that they have no conflict of interest.

# Acknowledgements

We appreciate the constructive reviews provided by three anonymous reviewers and editor Hans-Balder Havenith. The authors acknowledge the support from the National Key R&D Program of China under Grant 2017YFB0504101 and Grant 2021YFB3901201.

# Financial support

This research has been supported by the National Key R&D Program of China under Grant 2017YFB0504101 and Grant 2021YFB3901201.

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
