# Peer review of "Using single remote sensing image to calculate the height of the"

_Natural Hazards and Earth System Sciences, 2022_

## Author Comment (AC5)

Dear Reviewer#2,

Thank you for reviewing our manuscript and your valuable advice. We all agree with you completely. Based on the comments and suggestions, we have made extensive revisions to the original manuscript. A point-by-point response is presented below:

• **The title is ambiguous as the calculations are not merely based on a satellite image.**

Response:

Thanks for noting this. We took this title because the only after-landslide data we used in our me procedure is the remote sensing image. Your comment is right, beside satellite image, we also used some before-landslide data including the DEM data, the rough angle of repose of the soil. We just highlight the remote sensing image in the title for three main reasons. **Firstly,** a timely satellite image has a most significant influence on our method. **Secondly**, compared to other before-landslide data, we think access to satellite images is much more difficult, while other data have rather a reliable global product such as SRTM DEM or access to a rough estimation such as the repose angle. The required data for our process is shown in the following figure. **Last,** before our paper, the methods that offer prediction of the height of the landslide dam and the maximum volume of the lake usually require more than two visible spectral remote sensing images or a radar remote sensing image. The simple requirement of the application of our procedure is a great improvement. So, we aim to address this point in our title.

**Figure 1 the data required for completing this prediction.**

| Data name | Whether timely | Access | Optional/necessary | Explain |
|---|---|---|---|---|
| Satellite image | Yes | Satellite or UAV | Necessary | In fact, this image is used to classify the land are in reality. Therefore, it is possible to make use of other kinds of satellite data to achieve this goal. |
| DEM | No | Historical data (SRTM DEM) | Necessary | As the elevation of the land surface does not vary greatly without geological disasters, so the |

| | | | | |
|---|---|---|---|---|
| | | | | historical data is used in this method. |
| Repose angle of soil | No | Historical recordings | Optional | This data is usually obtained through experiment in lab or field survey. If can not get this through historical recordings, it is also reasonable to use the data in other papers or cases. And the sensitive analysis is carried out to show the influence of this parameter. |
| Elevation of river bed | No | Historical recordings | Optional | This parameter can be obtained by historical recordings directly, but if it is missing we also provide a method to get a rough estimation. And the sensitive analysis is carried out to show the influence of this parameter. |

These four kinds of data are required to complete the prediction. And the characteristic of them is shown above. In the consideration of this, we make the title. Although we agree with your comment. As a better choice is not found based on our discussion, we may keep this title in our manuscript. However, we will clarify this point in our revised manuscript.

- **Introduction should explain these parameters and also their connection with risk assessment.**

Response:

Thank you for your advice. We will clarify this point in the revised manuscript.

- **Most of the paper explains the long procedure with very short discussion and rapid hazard assessment section.**

Response:

Thank you for your advice. We will clarify this point in the revised manuscript. And we will add another case of landslide dam to verify our method. The discussion section will also be expended by the analysis of relevant rapid hazard assessment.

- **Rapid hazard assessment section needs more explanation.**

Response:

Thank you for your advice. We will give more explanation to Rapid hazard assessment in the revised manuscript.

- **The section should be supplemented with a figure, explaining the complete process of determination of parameters.**

Response:

Thank you for your advice. The rough figure that explains the process of determination of parameters is showed as following.

[Figure]

**Picture 1 the process of determination of parameters**

These words in red means the input data. The characteristics of them is shown in **Figure 1 before**. And we will give a more formular figure in our revised manuscript.

- **May I suggest to include another landslide dam for validation purpose**

Response:

Thank you for your reasonable advice. **And we will add another landslide dam in the manuscript for validation.**

We will revise our manuscript based on your helpful comments after interactive discussion.

Thanks again.

Weijie Zou on behalf of all co-authors

---

## Author Response (AR2)

MS No.: nhess-2022-35

Dear Referees, We are sincerely grateful for your constructive comments and suggestions that, we believe, have helped us to make the manuscript clearer, more precise and robust. Following the recommendations, we revised the manuscript based on your reviewers' comments. We address you point-by-point all the answers and we will modify the manuscript accordingly.

**Response to review comments by Anonymous Referee #1 (R1)**

General comment:
I think that paper can be accepted, though I still have some doubts about the sufficiency of just one example (the 2018 Baige landslide dam) to validate the general applicability of the proposed method. However, I expect that it will give start for some discussion.

**Authors' response:**
**Dear reviewer #1:**
**Thank you for your positive feedback! And we take your suggestion into serious discussion.**

**In our way of validating the theory, we have used some medium data such as the real lake area in the image and the evolution of the riverbed in order to take the error under control and support the process we promote. We think that the factor that influences the reliability of the outcome most is the process that we decide the input parameters. Therefore, we have performed sensitive analysis on the input parameters and the image solution. We agree that using only one example is vulnerable, and thus, in the following work, the discussion based on more cases will be carried out to support our method. And we are looking forward to more scholars taking part in the discussion about this method.**
**And we have added another case, Hongshiyan landslide dam, for validation in the revised manuscript.**

**Response to review comments by Anonymous Referee #2**

General comment:

The paper introduced a method to calculate different parameters, required for risk assessment of landslide dams. The paper is presenting an interesting method. However, following points should be considered before its acceptance for publication.

**Authors' response:**

Thank you for reviewing our manuscript and your valuable advice. We all agree with you completely. Based on the comments and suggestions, we have made extensive revisions to the original manuscript. A point-by-point response is presented below:

Comment:

**The title is ambiguous as the calculations are not merely based on a satellite image.**

**Authors' response:**

Thanks for noting this. We took this title because the only after-landslide data we used in our me procedure is the remote sensing image. Your comment is right, beside satellite image, we also used some before-landslide data including the DEM data, the rough angle of repose of the soil. We just highlight the remote sensing image in the title for three main reasons. **Firstly,** a timely satellite image has a most significant influence on our method. **Secondly**, compared to other before-landslide data, we think access to satellite images is much more difficult, while other data have rather a reliable global product such as SRTM DEM or access to a rough estimation such as the repose angle. The required data for our process is shown in the following figure. **Last,** before our paper, the methods that offer prediction of the height of the landslide dam and the maximum volume of the lake usually require more than two visible spectral remote sensing images or a radar remote sensing image. The simple requirement of the application of our procedure is a great improvement. So, we aim to address this point in our title.

**Figure 1 the data required for completing this prediction.**

| Data name | Whether timely | Access | Optional/necessary | Explain |
|---|---|---|---|---|
| Satellite image | Yes | Satellite or UAV | Necessary | In fact, this image is used to classify the land are in |

| | | | | reality. Therefore, it is possible to make use of other kinds of satellite data to achieve this goal. |
|---|---|---|---|---|
| DEM | No | Historical data (SRTM DEM) | Necessary | As the elevation of the land surface does not vary greatly without geological disasters, so the historical data is used in this method. |
| Repose angle of soil | No | Historical recordings | Optional | This data is usually obtained through experiment in lab or field survey. If can not get this through historical recordings, it is also reasonable to use the data in other papers or cases. And the sensitive analysis is carried out to show the influence of this parameter. |
| Elevation of river bed | No | Historical recordings | Optional | This parameter can be obtained by historical recordings directly, but if it is missing we also provide a method to get a rough estimation. And the sensitive analysis is carried |

| | | | | out to show the influence of this parameter. |
|---|---|---|---|---|

These four kinds of data are required to complete the prediction. And the characteristic of them is shown above. In the consideration of this, we make the title. Although we agree with your comment. As a better choice is not found based on our discussion, we may keep this title in our manuscript. However, we will clarify this point in our revised manuscript.

Comment:

**Introduction should explain these parameters and also their connection with risk assessment.**

**Authors' response:**

Thank you for your advice. We will clarify this point in the revised manuscript.

Comment:

**Most of the paper explains the long procedure with very short discussion and rapid hazard assessment section.**

**Authors' response:**

Thank you for your advice. We will clarify this point in the revised manuscript. And we will add another case of landslide dam to verify our method. The discussion section will also be expended by the analysis of relevant rapid hazard assessment.

Comment:

**Rapid hazard assessment section needs more explanation.**

**Authors' response:**

Thank you for your advice. We will give more explanation to Rapid hazard assessment in the revised manuscript.

Comment:

**The section should be supplemented with a figure, explaining the complete process of determination of parameters.**

**Authots' response:**

Thank you for your advice. The rough figure that explains the process of determination of parameters is showed as following.

[Figure]

**Figure 1 the process of determination of parameters**

These words in red means the input data. The characteristics of them is shown in **Figure 1 before**. And we will give a more formular figure in our revised manuscript.

Comment:

**May I suggest to include another landslide dam for validation purpose**

Authors' response:

Thank you for your reasonable advice. And we will add another landslide dam in the manuscript for validation.

And we take Hongshiyan landslide dam as an example. The predicting result is shown as below, which suggests a strong consistency between them:

| Parameter | Measured data | The predicting outcome | Error |
|---|---|---|---|

| the lowest elevation of the dam top | $1222\,(m)$ | $1216.7(m)$ | $5.3(m)$ |
| the maximum of lake volume | $2.60 \times 10^8 (m^3)^*$ | $3.09 \times 10^8 (m^3)$ | $0.49 \times 10^8 (m^3)^*$ |

**Response to review comments by Anonymous Referee #3**

General comment:
The authors proposed a procedure for calculating the landslide dam height and barrier lake volume. I think the procedure can effectively support the quick decision-making regarding hazard mitigation. I suggest that the paper can be accepted. I also give some comments for the author to modify the manuscript.

Authors' response:
We thank you for your careful reading of the manuscript and for many constructive comments and suggestions, which were useful to improve the manuscript. Please be kind to check the attached document for our reply to your comments in detail.

General comment: Lines 71-76. I think this paragraph is not appropriate in Section Introduction. In the Introduction Section, it is necessary to emphasize the disadvantages of the current methods for parameter calculation of the landslide dam.

Authors' response:
Thank you for your suggestion. We will make some adjustments in this paragraph and give more introduction to current methods for obtaining geometry parameters of landslide dams.

General comment: Two landslide dams occurred in the same position in Baige village. Please clarify which landslide dam is the object of the study.

Authors' response:
Thank you for your advice. In fact, the case mentioned in our paper is the second Baige landslide dam.

In order to demonstrate the proposed procedure, the second Baige landslide is taken as example. The image used is a 0.8m resolution image from Beijing-1 which was taken on November 9, 2018 and the pre-landslide DEM we choose is SRTM V3 of 30m resolution which was taken in 2000.

General comment: Are there any other methods to calculate the parameters of landslide dam, which can be used as a comparative analysis.

**Authors' response:**
Thank you for your advice. In fact, other methods to obtain the parameters is mainly based on field survey. The procedure we proposed is distinguished from them as we just use one single image to make the prediction and do not require a field survey. And other prediction based on remote sensing is mainly carried out by UAVs, using sequential images, which we introduce in Line 62-70.
We will make corrections and make it more clear in revised manuscript.

Comment: Line 43 "Generally speaking" and Line 71 "What's more", I think these words are not suitable for using in academic paper.

**Authors' response:**
Thank you for your advice. We will make corrections in the revised manuscript.

Comment: Line 84 please revise. DEM, not Dem

**Authors' response:**
Thank you for your advice. We will check our paper and make corrections in the revised manuscript.

Comment: Please revise the sentence of Line 196. I suggest it can be modified as "The data can be found in…" or list the references in the Figure legend.

**Authors' response:**
Thank you for your advice. We will make corrections in the revised manuscript.

Comment: The texts in Figure 2, 6 and 9 are too small. Please revise them.

**Authors' response:**
Thank you for your advice. We will revise the figures in the revised manuscript.

Comment: Lines 383-384. References should not be included in the conclusion.

**Authors' response:**

Thank you for your advice. We will make corrections in the revised manuscript.